

# Surface equilibrium vapor pressure of organic nanoparticles measured from the Dynamic-aerosol-size Electrical Mobility Spectrometer

Ella Häkkinen[1], Huan Yang[1], Runlong Cai[1,2], and Juha Kangasluoma[1]

[1]Institute for Atmospheric and Earth System Research/Physics, Faculty of Science, University of Helsinki, Helsinki, 00014, Finland.
[2]Shanghai Key Laboratory of Atmospheric Particle Pollution and Prevention (LAP[3]), Department of Environmental Science & Engineering, Fudan University, 200438 Shanghai, China

**Correspondence:** Huan Yang (huan.yang@helsinki.fi)

**Abstract.** Aerosol particles undergo continuous changes in their chemical composition and physical properties throughout their lifecycles, leading to diverse climate and health impacts. In particular, organic nanoparticle's surface equilibrium vapor pressure stands as a critical factor for gas-particle partitioning and is pivotal for understanding the evolution of aerosol prop­erties. Herein, we present measurements of evaporation kinetics and surface equilibrium vapor pressures of a wide array of

laboratory-generated organic nanoparticles, employing the Dynamic-aerosol-size Electrical Mobility Spectrometer (DEMS) methodology, a recent advancement in aerosol process characterization. The DEMS methodology is founded on the princi­ple that the local velocity of a size-changing nanoparticle within a flow field has a one-to-one correspondence with its local size. Consequently, this approach can facilitate the in situ probing of rapid aerosol size-changing processes, by analyzing the trajectories of size-changing nanoparticles within the classification region of a differential mobility analyzer (DMA). We em­

ploy DEMS with a tandem DMA setup, where a heated sheath flow in the second DMA initiates particle evaporation in its classification region. Through analysis of the DEMS response and the underlying mechanism governing the evaporation pro­cess, we reconstruct temporal radius profiles of evaporating nanoparticles and derive their surface equilibrium vapor pressures across various temperatures. Our results demonstrate a good agreement between the vapor pressures deduced from DEMS measurements and those documented in literature. We discuss the measurable vapor pressure range achievable with DEMS and

elucidate associated uncertainties. Furthermore, we outline prospective directions for refining this methodology and anticipate its potential to contribute to the characterization of aerosol-related kinetic processes with currently unknown mechanisms.

## 1  Introduction

Aerosol particles have a wide range of impacts, including their influence on Earth's climate and air quality (Kazil et al., 2010; Makkonen et al., 2012), reduction in visibility (Bäumer et al., 2008), and harmful effects on human health (Lelieveld et al.,

2015; Likhvar et al., 2015). Throughout their lifecycle, aerosols undergo continuous changes in their chemical composition and physical properties, resulting in varied climate and health effects. Therefore, understanding the evolution of aerosol properties



in processes like gas-phase synthesis (Artelt et al., 2003; Chen et al., 2018), atmospheric new particle formation (Kulmala et al., 2013; Zhang et al., 2012), and their impact on human airways (Frederix et al., 2018; Asgari et al., 2021) is crucial.

Aerosol size distribution is one of the most important properties in various aerosol formation and growth processes. Con-
ventionally, the evolution of aerosol size distributions is characterized by the Electrical Mobility Spectrometer (EMS) which comprises an aerosol charger, a mobility characterization module using the Differential Mobility Analyzer (DMA), and a particle number counting module. Since its inception (Knutson and Whitby, 1975), significant prior research has advanced the refinement and development of the setup, aiming for improved theoretical descriptions (Stolzenburg, 1988; Stolzenburg and McMurry, 2008; Stolzenburg, 2018), measurement of different aerosol processes (McMurry, 2000; Jen et al., 2014, 2015, 2016;
Wang et al., 2019, 2023), wider measurement size ranges (Kangasluoma et al., 2020; Jiang et al., 2011a, b), and faster scanning times (Wang and Flagan, 1990; Kulkarni and Wang, 2006a, b; Wang et al., 2018). During a measurement cycle, the voltage applied to the DMA can be adjusted incrementally to obtain the distribution of particle number relative to electrical mobility. An important prerequisite for the success of EMS measurements lies in ensuring that the timescale of aerosol size distribution evolution significantly exceeds that of the EMS voltage scanning cycle. This condition is crucial for maintaining a consistent
aerosol size distribution throughout the measurement. Consequently, conventional EMSs are not suitable for characterizing aerosol systems undergoing evolution on shorter timescales compared to the EMS voltage scanning cycle.

As an extension of conventional EMSs, we have recently introduced an alternative approach known as the Dynamic-aerosol-size Electrical Mobility Spectrometer (DEMS) (Yang et al., 2022, 2023a). DEMS is designed to conduct in-situ measurements of rapidly evolving aerosol systems, occurring at a timescale close to that of one EMS voltage scanning cycle. This is possible
because the particle inertia is negligible, and hence the particle trajectory in the sheath flow contains a complete set of information about the particle size change history. Building upon this foundation, the DEMS approach first analyzes the trajectories of particles undergoing size changes within the DMA classification region, guided by the response of a voltage scanning cycle. Subsequently, the insights derived from the particle trajectories are employed to faithfully reproduce the historical changes in particle size. This working principle could empower DEMS to undertake potential measurements of various sub-second
scale processes to probe very fast aerosol size changes. Such processes include condensation growth/evaporation shrinkage of aerosols triggered by rapid environmental changes or even the more rapid deliquescence and efflorescence phenomena.

In our previous works, we presented a complete theoretical framework describing particle transmissions within the classification region of the DEMS. Additionally, we demonstrated the methodology's capability to recover the historical changes in particle size profiles, exemplified through the aerosol evaporation process (Yang et al., 2022, 2023a). Traditionally, aerosol
evaporation processes are characterized using a tandem differential mobility analyzer (TDMA) (Rader and McMurry, 1986; Tao and McMurry, 1989). This method incorporates two DMAs positioned before and after an evaporation oven. The first DMA selects particles of known size, while the second one measures the particle size distribution after passing through the evaporation oven. The TDMA setup has also been untilized effectively to measure aerosol hygroscopic properties (Lei et al., 2020, 2022). Our DEMS setup parallels the TDMA method, with a major difference – instead of utilizing an evaporation oven,
we facilitate direct particle evaporation within the classification of the second DMA by elevating its sheath flow temperature. Consequently, rapid evaporation processes are measured and analyzed in-situ within the second DMA, while the first DMA



consistently selects and introduces monodispersed particles with known initial sizes. Similar setup was also used by Wright et al. (2016). Hogan's group has also conducted nice studies involving size changes within the DMA classification region (Ahonen et al., 2019; Li and Hogan, 2017; Oberreit et al., 2015). Their research predominantly explores molecular clusters, whereas the current DEMS method primarily targets nanoparticles in the measurements and analysis.

The DEMS method could lead to more reliable results because of the well-defined evaporation period being measured. This is due to the fact that only the evaporation occurring within the DMA classification region influences the output signal of a DEMS voltage scanning cycle. Consequently, no experimental bias will be introduced even if the evaporation process continues after particles exit the classification region. In our previous work, preliminary measurements on a type of organic particle of known evaporation rates have shown promising results for the DEMS approach. In spite of this, more experimental validations are needed to fully demonstrate the validity and robustness of this approach. Specifically, the applicable measurement conditions of DEMS, its sensitivity to various assumptions in the theoretical model, and its accuracy against experimental uncertainties need further elucidation. Example measurements are also needed to demonstrate how the DEMS approach can contribute useful data relevant to atmospheric aerosol studies.

To address these issues, this paper presents a comprehensive collection of experimental data using the DEMS methodology. The approach facilitates in-situ measurements of evaporation kinetics and vapor pressures for a broad spectrum of laboratory-generated organic nanoparticles. As main processes driving the evolution of aerosol particles, condensation and evaporation control the partitioning of compounds between the gas and the particle phase. In particular, the equilibrium vapor pressure of organic compounds is the key property governing these processes, and thus is crucial in understanding the evolution of aerosol particles (Pankow, 1994). We implement DEMS with a tandem DMA setup, where a heated sheath flow in the second DMA initiates particle evaporation in the classification region. To demonstrate DEMS performance, we reconstruct temporal radius profiles of the evaporating particles, deduce equilibrium vapor pressure from the DEMS response using our recently derived theoretical model, and compare the resulting values with literature values. Furthermore, we discuss the applicable measurement conditions for the DEMS, uncertainties associated with the resulted vapor pressures, and propose directions for improvement alongside potential future applications.

## 2 Methods

### 2.1 Laboratory experiments

In order to perform laboratory experiments to demonstrate the operation of the DEMS, we constructed a tandem DMA setup shown in Fig. 1. We used five different sample compounds: tetraethylene glycol (PEG4), pentaethylene glycol (PEG5), hexaethylene glycol (PEG6), dibutyl sebacate and glycerol. These compounds were chosen based on their wide and suitable vapor pressure range (from 0.00006 to 0.01 Pa at 295 K) and available literature data on their thermodynamic and kinetic parameters. The main properties of the sample compounds are summarized in Table 1. During measurements, the liquid sample was placed inside a tube furnace in a ceramic boat to generate sample particles by evaporation-condensation technique using nitrogen as a carrier gas. After generation, particles were charged with a radioactive charger (Ni-63 370 MBq), and introduced to the 1st



DMA, which was operated at room temperature (~295 K) with sheath flow rate of 20 L min$^{-1}$. Particles were assummed to be in equilibrium with the surroundings at the room temperature, and thereby evaporation were completely neglected before particles enter the 2$^{nd}$ DMA (note that PEG4 and glycerol are more volatile compared to other chemicals studied, and their evaporations were not neglected at room temperature and were treated with a modified theoretical framework; see Supporting Information Sect. 1). Nearly monodispersed particles with a radius of 125 nm were selected by the 1$^{st}$ DMA and directed to the

2$^{nd}$ DMA, which worked as the DEMS. The 2$^{nd}$ DMA was placed inside a heated chamber furnace to heat its sheath flow and initiate particle evaporation inside the classification region. The temperature inside the chamber furnace (i.e., temperature of the sheath flow of the 2$^{nd}$ DMA) was varied between 295 and 343 K, depending on the evaporated material. The tubing between the 1$^{st}$ DMA outlet and the 2$^{nd}$ DMA inlet was partly inside the chamber furnace, causing a slight heating effect/evaporation on the sample aerosol before entering to the 2$^{nd}$ DMA. This is expected to introduce slight uncertainty in the measurement

and is discussed in Sect. 3.3.3. The aerosol flow rate of the 2$^{nd}$ DMA was 1 L min$^{-1}$, and the sheath flow rate was varied between 10 and 20 L min$^{-1}$, corresponding to a residence time of 1.17 s to 2.23 s inside the DMA. Nitrogen was used as a carrier gas in the sample flow, while clean air was used in the sheath flow. As the sample to sheath flow ratio was relatively small, the gas surrounding the evaporating particles was treated as air in the model analysis. To perform the DEMS voltage scan, the voltage applied to the 2$^{nd}$ DMA was scanned in steps, and the particle number concentration was measured with a

condensation particle counter (CPC; Airmodus A20) with an inlet flow rate of 1 L min$^{-1}$. The voltage scan of each sample compound at different measurement conditions was repeated several times. The particle concentration was kept below 100 cm$^{-3}$ to minimize the risk of the evaporated vapors interfering the particle evaporation. A coiled tubing at the sheath inlet of the 2$^{nd}$ DMA ensured that the flowing sheath air was heated to the set furnace temperature before entering the sheath region. To monitor the temperature in the chamber furnace, a temperature sensor was mounted on the surface of the 2$^{nd}$ DMA. Both

DMAs are Hauke type (Winklmayr et al., 1991) with inner radius of 2.5 cm, outer radius of 3.3 cm, and length of 28 cm, and they were calibrated before the experiment at room temperature using 100 nm polystyrene latex (PSL) particles.

**Table 1.** Summary of the vapor properties of the compounds used in this study (T = 298.15 K).

| | Glycerol | PEG4 | PEG5 | PEG6 | Dibutyl sebacate |
|---|---|---|---|---|---|
| **Chemical formula** | $C_3H_8O_3$ | $C_8H_{18}O_5$ | $C_{10}H_{22}O_6$ | $C_{12}H_{26}O_7$ | $C_{18}H_{34}O_4$ |
| **Molar mass (g mol$^{-1}$)** | 92.09 | 194.23 | 238.28 | 282.33 | 314.46 |
| **Density (kg m$^{-3}$)** | 1250 (Sigma-Aldrich) | 1125 (Sigma-Aldrich) | 1126 (Sigma-Aldrich) | 1127 (Sigma-Aldrich) | 940 (Sigma-Aldrich) |
| **Diffusion coefficient (m$^2$ s$^{-1}$)** | $7.63 \times 10^{-6}$ (Bird et al., 2007) | $5.2 \times 10^{-6}$ (Krieger et al., 2018) | $4.66 \times 10^{-6}$ (Krieger et al., 2018) | $4.26 \times 10^{-6}$ (Krieger et al., 2018) | $3.2 \times 10^{-6}$ (Ray et al., 1979; Tang et al., 2015) |
| **Molecular volume (m$^3$)*** | $1.21 \times 10^{-28}$ | $2.93 \times 10^{-28}$ | $3.52 \times 10^{-28}$ | $4.16 \times 10^{-28}$ | $5.56 \times 10^{-28}$ |
| **Surface tension (N m$^{-1}$)** | 0.07 (Wright et al., 2016) | 0.045 (Gallaugher, 1932) | 0.045 (Gallaugher, 1932) | 0.045 (Gallaugher, 1932) | 0.04 (Zelko et al., 2002) |

*Calculated using density and molar mass of the compound.



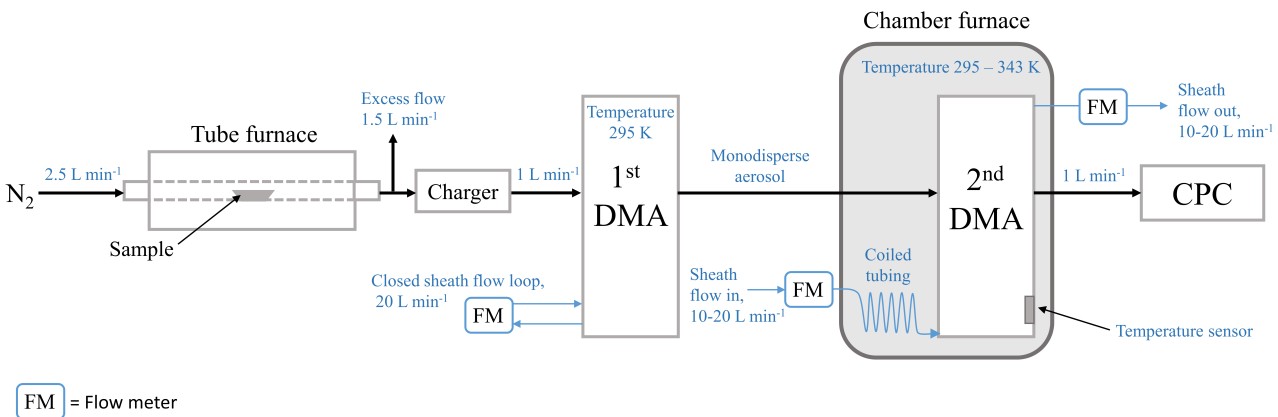

**Figure 1.** Schematic of the constructed tandem DMA setup.





## 2.2 Theoretical model

While the full theory has been presented in our previous paper (Yang et al., 2023a), a brief summary of the main idea of the theoretical model is given in this section for the completeness of presentation.

**Nanoparticle Time Average Mobility in the Second DMA.** By analyzing the equation of motion of inertialess nanoparticles moving within the electrical and flow fields inside the second DMA classification region, the "compressed information" carried by the "DEMS voltage scan" can be derived (Yang et al., 2023a):

$$\overline{Z_p} \equiv \frac{1}{t_f} \int\limits_0^{t_f} Z_p(t)dt = \frac{(2Q_{sh} + Q_a - Q_s)ln(r_2/r_1)}{4\pi L V_c}, \tag{1}$$

where $\overline{Z_p}$ is the time average mobility (hereafter termed as nominal mobility) of nanoparticles passing through the DMA, $t_f$ the nanoparticle residence time in the DMA classification region, $L$ the length of the DMA classification region, $V_c$ the centroid voltage, $r_2/r_1$ the outer shell/center rod radius of the DMA, $Q_a$ the aerosol flow rate, $Q_s$ the sampling flow rate, and $Q_{sh}$ the sheath flow rate. Eq. 1 states that the nominal mobility $\overline{Z_p}$ of the nanoparticle passing through the DMA is a constant, decided by the DMA geometrical parameters, flowrates, and centroid voltage. Though instructive, this equation alone cannot give the nanoparticle mobility change history, i.e., the expression of $Z_p(t)$.

**Time Dependent Mobility of an Evaporating Nanoparticle.** We then seek to provide the additional information needed to derive the expression of the time dependent mobility $Z_p(t)$, by examining the evaporation process that drives the nanoparticle size change. Here, nanoparticles are modelled as stagnant spherical droplets, and hence their time dependent mobilities/radii should follow some specific functional form according to the evaporation mass transfer regime; expressions for $Z_p(t)$ across the whole mass transfer regimes have been derived in our previous paper (Yang et al., 2023a). Nanoparticles used in this study (around 125 nm in radius) fall into the transition regime, so we opt to use the Transition regime (A) model in our previous paper (Yang et al., 2023a):

$$Z_p(t) \approx \frac{1}{\left(Ct + Z_{pi}^{-2}\right)^{\frac{1}{2}}}, \tag{2}$$

where $Z_{pi}$ is the mobility of the nanoparticle at the inlet slit of the second DMA classification region and $C$ a lumped constant depending on the related kinetic and thermodynamic parameters such as diffusion coefficient, surface tension, surface equilibrium vapor pressure.

An "approximately equal sign" is used in Eq. 2 because several assumptions have been made, including: (a) the evaporation Knudsen number in the transition regime evaporation model is assumed to be a constant, (b) the drag Knudsen number in the transition regime radius-mobility relationship is assumed to be a constant, and (c) the curvature of the nanoparticle (and hence the Kelvin effect correction to surface equilibrium vapor pressure) is also assumed to be a constant. All these constants are evaluated using the nanoparticle nominal/time average radius during evaporation. In reality, the above three quantities are



all time dependent as the nanoparticle radius decreases during evaporation. These assumptions, however, introduce negligible bias for the size range measured in this paper, which has been confirmed by comparing with numerical solutions where radius-dependent Kelvin effect and Knudsen number corrections are fully considered (see Sect. 3.2 and Sect. 3.3.2 for comparisons between theoretical model predictions and numerical simulation predictions).

Furthermore, for larger nanoparticles that fall into the diffusive regime, the evaporation rate and radius-mobility relationship are no longer sensitive to the Knudsen number, and the Kelvin effect becomes fully negligible. Under such condition, the right hand side of Eq. 2 is then a strict solution for the time-dependent mobility of an evaporating nanoparticle.

**Mobility and Radius at the Second DMA Outlet Slit.** It is noteworthy that lumping all thermodynamic and kinetic parameters into a single constant $C$ in Eq. 2 allows us to determine the nanoparticle's mobility change history and its mobility at the second DMA outlet slit from a single DEMS voltage scan. By substituting Eq. 2 into Eq. 1, the lumped constant $C$ can be obtained and expressed in terms of the nanoparticle nominal mobility, mobility at the inlet slit of the second DMA, and residence time:

$$C = \frac{\left[\frac{8\pi V_c L}{(2Q_{sh}+Q_a-Q_s)ln(r_2/r_1)} - Z_{pi}^{-1}\right]^2 - Z_{pi}^{-2}}{t_f} = \frac{\left(2\overline{Z_p}^{-1} - Zpi^{-1}\right)^2 - Z_{pi}^{-2}}{t_f} \tag{3}$$

Further, by substituting Eq. 3 into Eq. 2 and letting $t = t_f$, the mobility of the evaporating nanoparticle at the second DMA outlet slit $Z_{po}$ can be obtained:

$$Z_{po} = Z_p(t_f) = \left[\frac{8\pi V_c L}{(2Q_{sh}+Q_a-Q_s)ln(r_2/r_1)} - Z_{pi}^{-1}\right]^{-1} = \frac{1}{\left(2\overline{Z_p}^{-1} - Z_{pi}^{-1}\right)} \tag{4}$$

Nanoparticle mobilities at the DMA inlet and outlet slit can be converted to the corresponding nanoparticle radii based on the transition regime mobility-radius relationship derived from the corresponding transition regime drag model of a spherical nanoparticle:

$$Z_p = \frac{ie}{6\pi\mu r_p} \cdot \zeta(K_{ng}), \tag{5a}$$

where $i$ is the number of charges on the nanoparticle, $e$ is the elementary charge, $\mu$ is the background gas viscosity, and:

$$\zeta(K_{ng}) = 1 + K_{ng}\left(1.257 + 0.4exp\left(-\frac{1.1}{K_{ng}}\right)\right), \tag{5b}$$

is the Cunningham correction factor (Cunningham, 1910; Davies, 1945) with $K_{ng} = \lambda_g/r_p$ being the drag Knudsen number ($\lambda_g$ is the gas mean free path). In this and our previous work, the mobility-radius conversions have been done numerically using the above equations.

**Substance Flat Surface Equilibrium Vapor Pressure.** Once the nanoparticle radii at the DMA inlet and outlet slits are converted from the corresponding mobilities, the flat surface equilibrium vapor pressure $P_{ef}$ of the substance can be obtained based on the stagnant spherical droplet evaporation rate model (see equations 9 and 10 in our previous paper Yang et al. (2022)):




$$P_{ef} \approx \frac{k_B T(r_{pi}^2 - r_{po}^2)}{2D_v v_m exp\left(\frac{2\gamma v_m}{r_p k_B T}\right) f(\overline{K_{nv}}) t_f},$$  (6)

where $k_B$ is the Boltzmann constant, $T$ the temperature in the DMA classification region, $r_{pi}/r_{po}$ the nanoparticle radius at the second DMA inlet/outlet slit, $D_v$ the vapor diffusion coefficient, $v_m$ the vapor molecular volume, $\gamma$ the nanoparticle surface tension, and

$$f(K_{nv}) = \frac{(1 + K_{nv})}{1 + \left(\frac{4}{3\alpha} + 0.377\right) K_{nv} + \frac{4}{3\alpha} K_{nv}^2},$$  (7)

the Knudsen number dependent correction function (Li and Davis, 1996) ($\alpha = 1$ is the evaporation coefficient) with $K_{nv} =$

$\lambda_v / r_p$ being the evaporation Knudsen number ($\lambda_v$ is the vapor mean free path). $\overline{K_{nv}}$ means that it is evaluated using the nominal radius $\overline{r_p}$ converted from the nominal mobility $\overline{Z_p}$. Note that Eq. 6 directly gives the flat surface equilibrium vapor pressure of the substance composing the nanoparticle as Kelvin effect has been accounted for by the exponential term in the denominator. The "approximately equal sign" in Eq. 6 is used for a similar reason as that in Eq. 2, i.e., the Kelvin effect correction to the curved surface equilibrium vapor pressure and the Knudsen number correction to the transition regime evaporation rate are

assumed to be constants evaluated using the particle nominal radius. The diffusion coefficient is temperature dependent and its value for different temperatures can be estimated using (Tang et al., 2014):

$$D_v(298) = D_v(T) \cdot \left(\frac{298}{T}\right)^{1.75},$$  (8)

where $D_v(298)$ is the literature value for the diffusion coefficient at 298 K, and $D_v(T)$ is the diffusion coefficient at given temperature.

For convenience purpose, the term "vapor pressure" are used exchangeably with the "substance flat surface equilibrium vapor pressure" throughout the paper.

## 2.3 Numerical simulation

The trajectory of evaporating particles moving in the second DMA's classification region under the flow and electrical field are simulated as a verification of the theoretical model. Simulation details have been described in our previous paper (Yang et al.,

2023a). In the simulation, particles are treated in a Lagrangian perspective coupled to simplified Eulerian electrical and flow fields which are both assumed to be steady and uniform. The rate of evaporation for moving particles in the classification region is assumed to be equal to the evaporation of stagnant spherical droplets (the same treatment made in the theoretical model). We note that there is a radial velocity difference between the moving particle and the sheath air inside the DMA, but this velocity difference is small enough to be neglected in the model. At the beginning of the simulation, particles are released at the inlet

slit of the DMA. Their sizes are iterated every time step based on the stagnant spherical droplet evaporation rate model, and





trajectories are iterated by numerically solving the equation of motion resulted from the drag force (calculated from the size at the present time step) and the electrostatic force. For a given voltage, the portion of particles that can pass through the outlet slit of the DMA are recorded. Therefore, by varying the voltage applied to the DMA incrementally, the simulation results in the response of the DEMS, i.e., the "voltage-portion of particles passing through the DMA" curve. The centroid/peak voltage

of the response curve is identified and converted to nominal mobility and radius based on Eq. 1, Eq. 5a, and Eq. 5b. With other thermodynamic parameters fixed, the flat surface vapor pressure that leads to the match between the theoretical particle radius at the DMA outlet and that of the simulated radius is recorded. In this numerical simulation procedure, the time dependent effect of the evaporation Knudsen number, drag Knudsen number, and particle curvature are fully simulated. Hence, the vapor pressure values obtained from these simulations can serve as good verifications for the vapor pressure values calculated from

the theoretical approach in Sect. 2.2, where the three quantities are assumed to be time independent during evaporation.

## 3   Results and discussion

### 3.1   Response of the DEMS

The DEMS response curve of the transmission of evaporating particles can be obtained by scanning the voltage applied to the 2nd DMA. Example response curves for the PEG5 particle evaporation measured at different residence times and temperatures

are shown in Fig. 2. Before obtaining these curves, we performed benchmark experiments where both DMAs were operating at room temperature (295 K). Results confirm that the evaporation of PEG5 particles is negligible at room temperature. In addition, the 125 nm particles selected with the 1st DMA cannot be perfectly monodispersed (Knutson and Whitby, 1975), leading to slight broadening of the measured response curves. No special attention was paid to this broadening effect as it is not expected to influence the position of the centroid voltage on the response curve. We fit a Gaussian model to the response

curve to obtain the centroid voltage and convert it to nominal radius, i.e., the average particle radius during evaporation in the 2nd DMA.

Figs. 2 a-d show the DEMS response curves when the particle residence time in the 2nd DMA is kept constant while the temperature is increased from 298 K to 328 K. In general, a clear trend can be observed that the nominal radius of the evaporating particle decreases with the increase of temperature. At 298 K, the PEG5 particle radius decreases only slightly

from the initial radius of 125 nm, while the particle shrinkage becomes much more significant at the highest temperature of 328 K. Measurements were also performed on other compounds listed in Table 1. Benchmark experiments suggested that the evaporation of PEG5, PEG6, and Dibutyl sebacate can be neglect at room temperature, but that of glycerol and PEG4 cannot be neglected as they both shrank more than 6% in radius at room temperature (Fig. S4 and S5). We have extended the theoretical and numerical model to treat non-negligible room temperature evaporation in the 1st DMA and the tubing

between the two DMAs, when deducing the particle shrinkage and vapor pressure for glycerol and PEG4 (see Supporting Information for theoretical details). Fig. S1 shows the results of particle nominal radius against the evaporation temperature for all measured compounds. Clearly, glycerol has the highest volatility as it evaporates to a significantly smaller size compared to other compounds at the same temperature, whereas the PEG6 particle shrinkage is minor because of its lowest volatility.



We measured glycerol and PEG4 at two temperatures only, as higher temperatures could lead to complete evaporation (i.e.,
size decreasing to zero) of particles before reaching the outlet slit of the 2$^{nd}$ DMA. The complete evaporation of particles is
characterized by an additional peak/tail observed in the DEMS response (see Figs. S4, S5), which sets an upper limit for the
measurable vapor pressure range of the DEMS method and is discussed in greater detail in Sect. 3.3.2. Figs. 2 e-h show the
DEMS response curves of the evaporating particles when the temperature is kept constant while the residence time is increased
from 1.17 s to 2.23 s. Again, a clear trend can be observed that the measured nominal radius decreases with the increase of
particle residence time. Performing measurements at the same evaporation temperature but different particle residence times
allows us to reconstruct the particle size-variation history during evaporation. This point is discussed in more detail in the next
section.

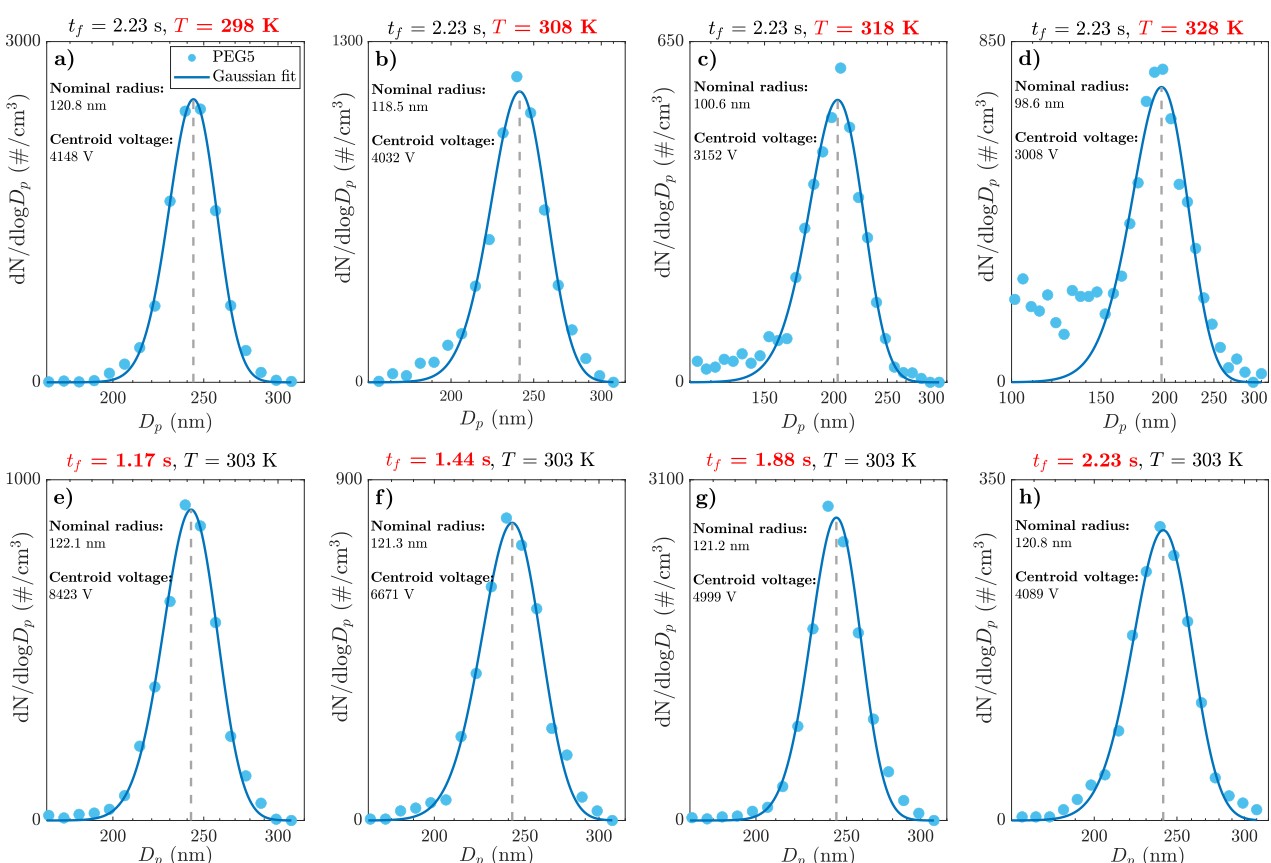

**Figure 2.** a,b,c,d) PEG5 particle evaporation at different DEMS temperatures and residence time of 2.23 s. e,f,g,h) PEG5 particle evaporation
at different DEMS residence times and temperature of 303 K.





## 3.2 Particle size-variation history

To demonstrate the process of reconstructing the particle size-variation history, we first measured the sample particles at a
constant temperature while changing the particle residence time inside the DEMS classification region (i.e. the sheath flow of
the 2$^{nd}$ DMA). We then converted the measured centroid voltages to nominal mobilities/radii using Eq. 1, and calculated the
particle mobilities/radii at the outlet of the 2$^{nd}$ DMA using Eq. 4. The resulting temporal evaporation profiles of the sampled
particles at any specified temperatures were hence re-constructed and shown in Figs. 3, S2, S3, S4 and S5. Further, vapor
pressure values at a specific measurement temperature but different residence times were calculated individually with Eq. 6
using the thermodynamic/kinetic parameters from Table 1, and the average of these individual vapor pressure values was taken
as the final theoretical vapor pressure for this specific temperature.

In order to verify the performance of the theoretical model, we performed numerical simulations as described in Sect. 2.3.
The simulated vapour pressure value is obtained by adjusting its value until the simulated particle outlet radius profile agrees
with the experimentally inferred one. In this context, it is worth noting that the particle's outlet radii derived from our theo-
retical model do not depend on precise values of the thermodynamic or kinetic parameters associated with the compound, as
illustrated in Eq. 1 and Eq. 4. However, when comparing measurements with numerical simulations to obtain the particle's out-
let radii, these thermodynamic or kinetic parameters of the compound become essential. This difference underlines a significant
advantage of our theoretical model. On the other hand, the thermodynamic and kinetic parameters are essential in the deter-
mination of vapor pressure values. Such a requirement remains unchanged whether one derives vapor pressure values through
a theoretical model or by means of comparisons with numerical simulations, and should hold true for any methods aimed
at deducing vapor pressure from the analysis of particle evaporation process. The vapor pressures obtained from the DEMS
theoretical model and the numerical simulation for each measured compound at different temperatures are listed in Table 2
together with available literature values. We found a good agreement between the theoretical and the simulated vapor pressure
values, suggesting that the assumptions made in Eq. 2 do not affect the accuracy of our theoretical model in reproducing the
size-variation history of evaporating particles.







**Figure 3.** PEG5 particle radius as a function of residence time inside the DEMS at five different temperatures. The particle inlet radius is 125 nm. Empty circles are the DEMS measured nominal radii, filled circles are radii at the outlet of the DEMS inferred from the measured nominal radii using the theoretical model. The red lines show the simulated nominal and outlet radii.





**Table 2.** Flat surface vapor pressures obtained from the DEMS theoretical model, the numerical simulation and literature for each sample compound at different temperatures.

| Compound | Temperature (K) | DEMS theoretical vapor pressure (Pa) | DEMS simulated vapor pressure (Pa) | Literature value for vapor pressure (Pa) |
|---|---|---|---|---|
| **Glycerol** | 295.15 | $0.0063^{+0.0016}_{-0.0010}$ | $0.0060^{+0.0015}_{-0.0010}$ | 0.0150 (CHERIC) |
| | 303.15 | $0.0093^{+0.0024}_{-0.0016}$ | $0.0095^{+0.0024}_{-0.0016}$ | 0.0398 (CHERIC) |
| **PEG4** | 295.15 | $0.0014^{+0.00040}_{-0.00022}$ | $0.0015^{+0.00040}_{-0.00025}$ | 0.0167 (Krieger et al., 2018) |
| | 303.15 | $0.0023^{+0.00060}_{-0.00040}$ | $0.0023^{+0.00057}_{-0.00030}$ | 0.0203 (Krieger et al., 2018) |
| **PEG5** | 298.15 | $0.00096^{+0.00024}_{-0.00018}$ | $0.0011^{+0.00026}_{-0.00019}$ | 0.0006 (Krieger et al., 2018) |
| | 303.15 | $0.0012^{+0.00030}_{-0.00022}$ | $0.0013^{+0.00030}_{-0.00025}$ | 0.000758 (Krieger et al., 2018) |
| | 308.15 | $0.0012^{+0.00030}_{-0.00020}$ | $0.0014^{+0.00035}_{-0.00023}$ | 0.00129 (Krieger et al., 2018) |
| | 318.15 | $0.0068^{+0.0017}_{-0.0011}$ | $0.0068^{+0.0017}_{-0.0012}$ | 0.00607 (Krieger et al., 2018) |
| | 328.15 | $0.0073^{+0.0018}_{-0.0012}$ | $0.0073^{+0.0019}_{-0.0013}$ | - |
| **PEG6** | 298.15 | $0.00054^{+0.00013}_{-0.00010}$ | $0.00067^{+0.00012}_{-0.00008}$ | 0.0000625 (Krieger et al., 2018) |
| | 313.15 | $0.00054^{+0.00013}_{-0.00010}$ | $0.00070^{+0.00013}_{-0.00010}$ | 0.000312 (Krieger et al., 2018) |
| | 323.15 | $0.00096^{+0.00024}_{-0.00017}$ | $0.0011^{+0.00030}_{-0.00020}$ | 0.000875 (Krieger et al., 2018) |
| | 333.15 | $0.0012^{+0.00040}_{-0.00020}$ | $0.0014^{+0.00035}_{-0.00025}$ | - |
| | 343.15 | $0.0035^{+0.00090}_{-0.00060}$ | $0.0035^{+0.00093}_{-0.00055}$ | - |
| **Dibutyl sebacate** | 295.15 | $0.00027^{+0.000070}_{-0.000050}$ | $0.00043^{+0.00010}_{-0.000060}$ | 0.00032 (Ray et al., 1979), 0.00049 (Small et al., 1948) |
| | 300.15 | $0.0017^{+0.00050}_{-0.00020}$ | $0.0018^{+0.00045}_{-0.00030}$ | 0.00063 (Ray et al., 1979), 0.00094 (Small et al., 1948) |
| | 305.15 | $0.0026^{+0.00050}_{-0.00030}$ | $0.0028^{+0.00055}_{-0.00040}$ | 0.0012 (Ray et al., 1979), 0.0018 (Small et al., 1948) |
| | 310.15 | $0.0029^{+0.00070}_{-0.00050}$ | $0.0031^{+0.00055}_{-0.00040}$ | 0.0024 (Ray et al., 1979), 0.0033 (Small et al., 1948) |
| | 313.15 | $0.0038^{+0.0010}_{-0.00060}$ | $0.0039^{+0.0010}_{-0.00060}$ | 0.0035 (Ray et al., 1979), 0.0047 (Small et al., 1948) |
| | 318.15 | $0.0076^{+0.0019}_{-0.0013}$ | $0.0070^{+0.0020}_{-0.0013}$ | 0.0065 (Ray et al., 1979), 0.0083 (Small et al., 1948) |





### 3.3 Vapor pressure

#### 3.3.1 DEMS measurable vapor pressure range

A complete evaporation of particles in the classification region of the DEMS determines the maximum measurable vapor pressure, whereas the ability of the DMA to measure very small shrinkage of particles (DMA resolution) limits the minimum

measurable vapor pressure. Therefore, the vapor pressure range that can be measured with the DEMS depends on the used inlet particle radius, the residence time inside the DEMS, the properties of the measured compound, and the dimensions and resolution of the DMAs used. To have an idea of the measurable vapor pressure range of the DEMS, we evaluated the theoretical and simulated minimum and maximum vapor pressure values measurable for the DEMS setup in this study. The results for the DEMS operating at 295 K with residence time of 1.17 s (sheath flow rate of 20 LPM), and inlet particle radius of 125

nm and 220 nm are shown in Fig. 4 and Table S1. Maximum measurable vapor pressure values were obtained by assuming a situation where the particles have completely evaporated in the DEMS by setting their outlet radius as 0 nm. For the minimum measurable vapor pressure values, we estimated that the minimum decrease in particle radius that is measurable with the current setup is ~1 % of the inlet particle radius. This estimation was done based on the standard deviation of the measured nominal radii of replicate measurements.

As can be seen from Eq. 6, the measurable vapor pressure values increase as the inlet particle radius increases, but decrease as the residence time increases. Thus, when the aim is to measure compounds with high vapor pressures, the best combination is a largest possible inlet particle radius and a shortest possible residence time, and vice versa when aiming to measure compounds with low vapor pressures. In terms of compound properties, we found that the key properties influencing the process are the compound's diffusion coefficient and its molecular volume (Table 1). According to Eq. 6, the larger the product of these two

is, the smaller the measurable vapor pressures are. In general, the DEMS measurable vapor pressure spreads about two orders of magnitude near the range of about $10^{-3}$ to $10^{-1}$ Pa with the setup used in this study. This measurable vapor pressure range can be further extended by using other types of DMAs. For example, a half-mini DMA with a residence time of 0.0048 s (Cai et al., 2018) can potentially push the maximum measurable vapor pressure up to about 0.3 Pa (17 nm radius glycerol particle), three times larger than that of the current setup. It is worth noting that, in this context, prioritizing the extension of the

maximum measurable vapor pressure holds greater importance than extending the minimum measurable vapor pressure. This is because that the maximum measurable vapor pressure signifies the method's capacity to effectively capture rapid size-variation processes with shorter time scales. While in cases involving compounds with low vapor pressures, the evaporation proceeds over longer time scales, making it feasible to directly measuring the processes using the conventional TDMA setups, and hence invoking the DEMS method becomes less advantageous.

#### 3.3.2 Vapor pressures measured by DEMS and comparison with literature values

To evaluate the accuracy of the vapor pressure obtained from the DEMS, we compared our measured values with values found in literature. In the used literature data the Kelvin effect is negligible, and thus the values are comparable to the flat surface vapor pressures measured in this study. For the different PEG compounds, we used the reference data set provided



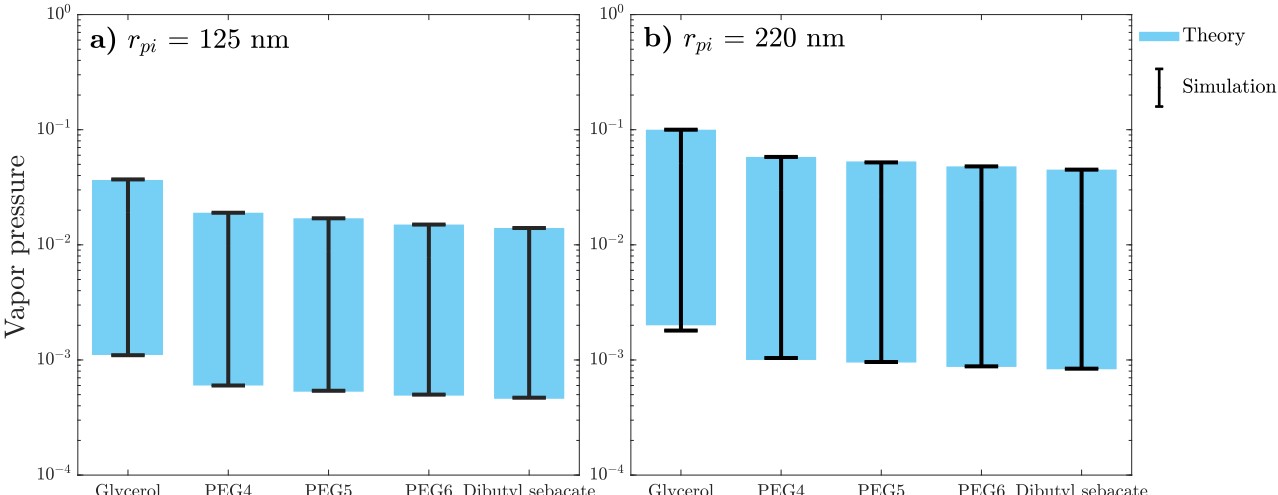

**Figure 4.** The DEMS maximum and minimum measurable vapor pressures at temperature of 295 K, residence time of 1.17 s and inlet particle radius of 125 and 220 nm.

by Krieger et al. (2018), where they measured PEG vapor pressures using three different techniques: particles levitated in an electrodynamic balance (EDB) to measure diffusion-controlled evaporation rates, the laminar flow tube–tandem differential mobility analyzer (FT-TDMA) to measure the particle size distribution before and after evaporation in a laminar flow tube, and the Knudsen effusion mass spectrometer (KEMS) to measure the concentration of the vapor effusing from a macroscopic sample in a Knudsen cell. For dibutyl sebacate we used the values determined from vapor effusion through a small orifice (Small et al., 1948) and from droplet evaporation (Ray et al., 1979). Glycerol vapor pressure literature values we found from the Chemical Engineering and Materials Research Information Center (CHERIC).

Figure 5 shows the vapor pressure values for PEG5, PEG6 and dibutyl sebacate measured in this study with the DEMS (both theoretical and simulated values) as well as the values reported in literature as a function of scaled inverse temperature. While there is generally a good agreement between the DEMS vapor pressure values and the literature values, it is noteworthy that the DEMS values tend to be slightly higher than the literature values. Higher vapor pressures measured by the DEMS are likely due to partial particle evaporation occurring in the 1st DMA and in the tubing between the 1st and 2nd DMA, resulting in an overestimation of evaporation rates in the 2nd DMA classification region. Moreover, vapor pressures obtained from the DEMS theoretical model are generally slightly lower than the vapor pressures obtained from the DEMS numerical simulation. This leads to a more favorable alignment between the theoretical vapor pressure values and the literature values; however, it is probable that this alignment is primarily an artifact of the assumptions made in the theoretical model. For PEG6, the vapor pressure measured at 298 K is notably higher than the literature values (Table 2). This is expected, as the literature value for PEG6 vapor pressure is 6.25 x $10^{-5}$ Pa at 298 K, whereas the PEG6 minimum measurable vapor pressure at this temperature



is higher, 5 x $10^{-4}$ Pa (Table S1). In our measurement the PEG6 particle nominal radius decreased about 1.6 % at 298 K, but to have a matching value with the literature it should have decreased 0.2 %, which is too small shrinkage to be accurately measured with the resolution of the DMA used in this study.

Glycerol and PEG4 are more volatile and have higher vapor pressures than the other three compounds used in this study. We observed that glycerol shrank about ~17 % and PEG4 shrank about ~7 % in nominal radius already at 295 K. These suggest non-negligible evaporation at room temperature for these two compounds. Thus, their evaporation in the $1^{st}$ DMA (operating at 295 K) was taken into account when deducing the vapor pressures (see Supporting Information for theoretical details). The vapor pressures obtained from the DEMS for these two compounds seem to be in a poor agreement with the literature values,
likely because their vapor pressures at these temperatures are close to the maximum measurable value (Table S1). As has been mentioned earlier, when the vapor pressure of the compound is close to or larger than the maximum measurable value, some of the particles may fully evaporate during the measurement, and a peak/tail appeared on the left of the DEMS response curve. These tails were found in the measurements of glycerol and PEG4 at 303 K (Fig. S4, Fig. S5).

**Figure 5.** Comparison of different vapor pressure values for a) PEG5, b) PEG6 and c) dibutyl sebacate measured with different methods. DEMS theoretical model and numerical simulation values are from this study and other values from literature.



### 3.3.3 Uncertainties in the vapor pressure measured by DEMS

Our approach of obtaining the vapor pressure comprises of two separate parts. In the first part, we calculate the particle radius at the outlet slit of the $2^{nd}$ DMA ($r_{po}$) from the centroid voltage of the DEMS measurement and the particle radius at the inlet slit of the $2^{nd}$ DMA ($r_{pi}$). In the second part, we use the values of $r_{po}$ and $r_{pi}$ to calculate the vapor pressure (Eq. 6) based on the evaporation rate model. The uncertainty of the calculated vapor pressure hence comes from both two parts.

We start with giving an estimation of the uncertainties in the calculated particle radius at the $2^{nd}$ DMA outlet ($r_{po}$). First,
Eq. 1 used in the theoretical model is expected to have negligible error, as it is derived by solving the equation of motion for nanoparticle trajectories in the $2^{nd}$ DMA classification region. Second, as has been noted earlier, in reaching Eq. 2, we assume that the Kelvin effect correction to the curved surface vapor pressure and the Knudsen number correction to the transition regime evaporation rate are both constants (calculated using the nanoparticle nominal radius during evaporation). However, by comparing with numerical simulations where radius-dependent Kelvin effect correction and Knudsen number correction are
used, the errors resulted from the above approximations have been found to be negligible for the studied size range (Yang et al., 2023a). Third, once the nanoparticle mobility at the $2^{nd}$ DMA outlet slit is calculated from combining Eqs. 1 and 2, it will be converted to the corresponding radius by the mobility-radius relationship. This step can also be assumed to have negligible error as the mobility-radius relationship have been calibrated with PSL particle size standards before the experiments. Last, a small portion of the tube connecting the $1^{st}$ and $2^{nd}$ DMA is inevitably placed inside the chamber furnace. Hence, before
entering the $2^{nd}$ DMA for evaporation measurement, particles may have already experienced partial evaporation. This effect is not considered in the theoretical and numerical model, leading to uncertainties in the calculated value of $r_{po}$. The length of this portion of tube inside the chamber furnace is 14 cm, leading to particle residence time of ~0.2 s. Based on numerical simulations of the evaporation process, we estimate that the actual particle shrinkage in the $2^{nd}$ DMA classification region is 2 % smaller than the shrinkage in ideal conditions (i.e., the particle shrinkage calculated using the theoretical model and the ideal
initial size of $r_{pi}$=125 nm selected by the $1^{st}$ DMA). However, it should be noted that this estimation applies only to the case of PEG5, PEG6, and dibutyl sebacate whose evaporation at room temperature can be completely neglected. For glycerol and PEG4 (more volatile), the corresponding error is expected to be much larger, even if the modified theoretical model provided in the Supporting Information was employed. Nevertheless, for the compounds shown in Fig. 5, the shrinkage rate is small at the beginning of the evaporation process, hence the decrease in particle radius in the first 0.2 s of the evaporation is negligible
and not considered when estimating uncertainties.

We then check the uncertainties associated with applying the evaporation rate model to calculate the vapor pressure (Eq. 6). Like the assumptions made in Eq. 2, those made in Eq. 6 (i.e., the Kelvin effect correction to the curved surface vapor pressure and the Knudsen number correction to the transition regime evaporation rate are constants evaluated using the particle nominal radius) have negligible error and hence will not lead to potential uncertainties. The main uncertainty in the vapor pressure
comes from the thermodynamic and kinetic parameters used in Eq. 6. Among these parameters, surface tension should have minimal influence because the Kelvin effect is not significant for the measured size range. Also molecular volume, which is calculated from molecular weight and density, has negligible effect on the uncertainty based on the minimal uncertainty in the





density measurements (Viana et al., 2002). Moreover, it should be safe to assume that the evaporation coefficient is close to 1; recent molecular dynamics simulations suggested that the evaporation coefficient only drops to less than 1 for extremely small

clusters (composed of several molecules) at temperatures approaching boiling point (Yang et al., 2023b). Finally, the calculated vapor pressure should be sensitive to the value of the diffusion coefficient used in Eq.6.

To estimate the uncertainty range, we set the uncertainty in diffusion coefficient to 20 % (Krieger et al., 2012; Huisman et al., 2013) in both theoretical calculations and numerical simulations. The resulted vapor pressure ranges are listed in Table 2 and marked as uncertainty bars in Fig. 5. The DEMS uncertainty bars largely overlap with the literature values, indicating a

good agreement between the values, and reinforcing the accuracy of the DEMS in measuring the flat surface vapor pressures of organic nanoparticles.

### 3.4   Future directions of improvement for DEMS

The analysis presented in this paper is limited to processes with well-established fundamental mechanisms. This limitation arises from the inherent capability of DEMS voltage scans, which can only provide the time-averaged mobility of nanoparticles

undergoing size changes. Therefore, additional information is essential to fully solve the nanoparticle size change history. In the context of nanoparticle evaporation, this additional information is derived from the known evaporation mechanism. This raises a pivotal question: can DEMS extend its capability to processes with entirely unknown mechanisms? As highlighted in our previous work, one potential way for achieving this is through measurements at different particle residence times, yielding a set of equations representing the Fredholm integrals of the first kind with a constant kernel function (see Eq. 46 in Yang et al.

(2023a) and the corresponding paragraph). These integral equations are readily solvable and can result in the nanoparticle size change history without prior knowledge of the underlying mechanism. This promising direction of refinement could render the method viable for exploring complex kinetic processes such as the hygroscopic behavior of nanoparticles, the transformation of multi-component nanoparticles, and the kinetic intricacies exhibited by exceptionally small nanoparticles where conventional bulk models fall short. These complex kinetic phenomena, encompassing evaporation, condensation, and water absorption,

play crucial roles in various atmospheric aerosol processes yet remain elusive. The versatility of our approach could potentially open doors to investigate previously uncharted territories. However, to comprehensively evaluate the reliability of our method for future exploration, dedicated numerical simulations are crucial to carefully assess uncertainties, in which the concentration, temperature, and flow fields need to be fully resolved. Simultaneously, concerted efforts are needed to refine the mechanical, thermal, and structural aspects of the experimental setup, ensuring the method's robustness for shorter residence times, faster

processes, and smaller nanoparticle sizes.

### 4   Conclusions

We demonstrated the recently developed DEMS methodology by conducting measurements on the evaporation kinetics and vapor pressures of diverse laboratory-generated organic nanoparticles across temperatures ranging from 295 to 343 K. Based on the DEMS responses and the underlying mechanism driving nanoparticle evaporation, we successfully reconstructed the



temporal radius profiles of evaporating particles, validating them against thorough numerical simulations. Moreover, we derived vapor pressures of the measured compounds at various temperatures from these reconstructed temporal evaporation profiles. Our findings reveal a robust agreement between the vapor pressures deduced from DEMS measurements and those reported in literature. Additionally, we assessed the uncertainty range and the measurable vapor pressure range of the DEMS approach. With the current setup, the measurable vapor pressure ranges from around $10^{-3}$ Pa to $10^{-1}$ Pa, but this range can be further

extended by selecting proper particle initial radius and adjusting the particle residence time in the DMA classification region.

The DEMS concept is founded on the principle that the local velocity of a size-changing nanoparticle in a flow and electrical field corresponds directly to its local size. This unique relationship enables the reconstruction of a nanoparticle's size-changing history based on its trajectory within the DMA classification region. This fundamental working principle presents significant potential for further enhancing the capability of the DEMS. Specifically, it is essential to develop a reliable theoretical model

for inversely deducing the nanoparticle trajectory within the DMA classification region from DEMS responses under various working conditions, while without relying on the specific mechanism of the size-changing process itself. Such advancement can expand the DEMS's utility to processes governed by entirely unknown mechanisms. Additionally, improvements in experimental design are crucial to minimize the residence time of nanoparticles within the DMA classification region. Such optimization can facilitate the DEMS in capturing more rapid processes that are not measurable using conventional approaches. Through

continued refinement and exploration, we anticipate that the DEMS approach will contribute to deepening our understanding of complex nanoparticle kinetic phenomena and enhancing its adaptability across diverse and challenging scenarios.

*Data availability.*    Data are available upon request from the authors.

*Author contributions.*    EH, HY, RC and JK designed the study. HY provided the theoretical model and performed numerical simulations. EH conducted the experiments. EH and HY analyzed the data and prepared the manuscript with contributions from all other authors.

*Competing interests.*    The authors declare that they have no conflict of interest.

*Acknowledgements.*    We are grateful for the funding from the Research council of Finland (356134, 346370, 325656) and the Vilho, Yrjö and Kalle Väisälä Foundation.





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
