# Peer review of "Surface equilibrium vapor pressure of organic nanoparticles measured from the Dynamic-aerosol-size Electrical Mobility Spectrometer"

_EGUsphere, 2024_

## Author Comment (AC1)

**Response to Reviewer #1**

**General comment:**

This paper develops a modified tandem DMA method to examine particle vapor pressures. Somewhat uniquely, size changes within a DMA are used to infer evaporation rates, and hence vapor pressures. Though the idea to use a DMA as a "reaction cell" of controlled residence time is not new, I think this is a novel contribution to the literature and I only have several minor suggestions and questions for the authors.

Thank you for your positive feedbacks. We appreciate your comments which help to improve our manuscript. We have made point-by-point answers to your comments below:

**Comment #1:**

The authors are essentially exploiting the fact the DMA is a controlled residence time instrument, with the residence time determined by the ratio of the sheath velocity to DMA length, and the resolution on the residence time equivalent to the DMA resolution. I think it is fine if the authors want to name the use of a DMA to monitor reaction kinetics through controlled residence time the "Dynamic-aerosol-size Electrical Mobility Spectrometer." However, I think they should note that there are other studies that have used this technique previously. I believe the most correct work from the Hogan group the authors should reference is:

Li, Chenxi, and Christopher J. Hogan Jr. "Direct observation of C 60− nano-ion gas phase ozonation via ion mobility-mass spectrometry." Physical Chemistry Chemical Physics 21.20 (2019): 10470-10476

This work uses the DMA to monitor reaction kinetics (the cited references are all focused on equilibrium vapor uptake. Fernandez de la Mora also utilized a nearly identical approach to the authors to look at evaporation of clusters:

Fernandez de la Mora et al. "Measuring the kinetics of neutral pair evaporation from cluster ions of ionic liquid in the drift region of a differential mobility analyzer." The Journal of Physical Chemistry A 124.12 (2020): 2483-2496.

Not using a DMA but a drift cell, the Clowers group has used mobility controlled residence times to look at hydrogen-deutrium exchange reactions in the gas phase:

Schramm, Haley M., et al. "Evaluation of Hydrogen–Deuterium Exchange during Transient Vapor Binding of MeOD with Model Peptide Systems Angiotensin II and Bradykinin." The Journal of Physical Chemistry A 127.42 (2023): 8849-8861

Schramm, Haley M., et al. "Ion-neutral clustering alters gas-phase hydrogen–deuterium exchange rates." Physical Chemistry Chemical Physics 25.6 (2023): 4959-4968.

**Response:**

Thank you for your notes. The works you mentioned are quite relevant to our study. We have incorporated a discussion in the introduction to highlight both the similarities and differences between these studies and our own. From an application perspective, the primary distinction lies in

the focus: the mentioned studies investigate molecular clusters/ions, whereas our research targets nanoparticles. On the theoretical side, our study differs in that the size variation of a nanoparticle is modelled as a continuous function. This continuity allows us to develop a rigorous particle transmission theory based on the original framework of Knutson and Whitby. In contrast, previous work on molecular clusters typically deals with size changes that involve discontinuities, and thus a similar theory has not been established in previous studies.

We revised the introduction section as follows: "Our DEMS setup parallels the TDMA method, with a major difference – instead of utilizing an evaporation oven, we facilitate direct particle evaporation within the classification of the second DMA by elevating its sheath flow temperature. Consequently, rapid evaporation processes are measured and analyzed in-situ within the second DMA, while the first DMA consistently selects and introduces monodispersed particles with known initial sizes. Similar setup was also used by Wright et al. (2016). Other groups have also conducted studies initiating evaporation or other processes in the DMA classification region. Fernandez de la Mora et al., 2020 studied cluster evaporation within the DMA, while Oberreit et al., 2015, Ahonen et al., 2019 and Li and Hogan, 2017 explored vapor uptake of clusters. Furthermore, studies investigating gas-phase reactions within an ion mobility system have been performed previously (Li and Hogan, 2019; Schramm et al. 2023a; Schramm et al. 2023b). These studies predominantly focus on ions and molecular clusters, whereas the current DEMS method primarily targets nanoparticles in the measurements and analysis."

**Comment #2:**

*Equation (7). There are other published equations for the transition regime condensation/evaporation equation provided in equation 7. How sensitive are results to this equation, in comparison to the equation of Dahneke or Gopalakrishnan & Hogan? Also, how is the vapor mean free path explicitly defined for this equation? Different references commonly give different definitions of this parameter (it needs to be proportional to the vapor diffusion coefficient divided by the vapor mean thermal speed, but the proportionality coefficient depends on the theory).*

**Response:**

Exactly, the vapor mean free path is proportional to the vapor diffusion coefficient divided by the vapor mean thermal speed. The vapor mean free path was explicitly defined in our work using the kinetic relationship: $\lambda_v = 3D_v/c_v$. The use of this relationship ensures that the transition regime evaporation rate converges to the continuum and free molecular regime solution when the evaporation Kn number approaches the lower and higher end (Eqs. 36a, 36b, 41a, and 41b in *Yang et al. 2023, 170, 106141, J. Aerosol Sci.*).

The results in this paper are not sensible to different forms of transition regime condensation/evaporation equations. This is a coincidence, because the initial sizes of the particles used in this study (125 nm) are comparatively large. Their evaporation largely falls into the continuum regime, and hence Kn number correction to the evaporation rates are not significant. Thanks for your notes; these points do need extra highlights in the text, and we have added the following texts:

Pg. 8: "Note that Eq. 6 directly gives the flat surface equilibrium vapor pressure of the substance composing the nanoparticle as Kelvin effect has been accounted for by the exponential term in the denominator. The "approximately equal sign" in Eq. 6 is used for a similar reason as that in Eq. 2,

i.e., the Kelvin effect correction to the curved surface equilibrium vapor pressure and the Knudsen number correction to the transition regime evaporation rate are assumed to be constants evaluated using the particle nominal radius. Moreover, the results in this paper are expected to be insensible to different forms of transition regime condensation/evaporation equations (Gopalakrishnan & Hogan, 2011). Because the initial sizes of the particles used in this study (125 nm) are comparatively large, their evaporation largely falls into the continuum regime, and hence Knv correction to the evaporation rates are not significant. The diffusion coefficient in Eq. 6 is temperature dependent and its value for different temperatures can be estimated using (Tang et al., 2014):"

Pg. 8: "where $D_V(298)$ is the literature value for the diffusion coefficient at 298 K, and $D_V(T)$ is the diffusion coefficient at given temperature. In addition, the vapor mean free path was explicitly defined in our work using the kinetic relationship: $\lambda_v = 3D_v/c_v$. The use of this relationship ensures that the transition regime evaporation rate converges to the continuum and free molecular regime solution when the evaporation Knv approaches the lower and higher end (see Eqs. 36a, 36b, 41a, and 41b in *Yang et al. 2023, 170, 106141, J. Aerosol Sci.*)."

**Comment #3:**

Table 2. Could the authors comment further on the disparity between the vapor pressures determined for glycerol and the literature values? Unlike the PEGs, the glycerol vapor pressure appears to be much lower by DEMS than reported elsewhere. The PEG4 DEMS results are also lower in vapor pressure than Kreiger et al.

**Response:**

Glycerol and PEG4 are more volatile than the other used sample compounds, and their sizes have already shrunk at room temperatures before entering the heated region of the DEMS (the 2nd DMA in the setup). In this case, the 1st DMA operating at the room temperature cannot give reliable nanoparticle initial size which is an essential input for the model. This leads to errors in the prediction. When both DMAs are at room temperature (~295 K): glycerol evaporated ~14 % and PEG4 evaporated ~6 % for a residence time of 1.17 s. In contrast for other compounds, the evaporation at room temperature was negligible (~1 %,).

To consider the significant evaporation of glycerol and PEG4 at the room temperature, we made an extension to the theoretical model (shown in the Supporting information) and calculated the vapor pressures for these two compounds using the extended model. However, due to extra assumptions introduced in the extended model, its accuracy is also decreased.

Moreover, the response curves for glycerol and PEG4 (Figs. S5 and S6) have additional peak/tail on the left side of the curve, indicating that some of the particles have completely evaporated during the measurements. We defined the maximum measurable vapor pressures for each compound based on a situation where the particles have completely evaporated in the DEMS by setting their outlet radius to 0 nm. By comparing the glycerol and PEG4 literature values (Table 2) to the maximum measurable values in Table S1 ($rp_i$ = 125 nm), we can see that it is operating at the upper measurable limit of the DEMS.

We revised the sections in the manuscript discussing the glycerol and PEG4 measurements to highlight the points mentioned above.

---

## Author Comment (AC2)

**Response to Reviewer #2**

**General comment:**

The authors present an application of the recently developed DEMS methodology in the aerosol evaporation kinetics measurement. Evaporation of laboratory-generated single-component organic particles, including Glycerol, PEG4, PEG5, PEG6, and Dibutyl sebacate were measured and investigated across temperatures ranging from 295 to 343 K. The vapor pressures deduced from DEMS measurements are generally in good agreement with literature values, demonstrating that the DEMS is capable of characterizing aerosol evaporation kinetics. This manuscript is well-written and fits the scope of Atmospheric Measurement Techniques. The reviewer recommends accepting this manuscript after addressing the following minor comments.

We thank the reviewer for the positive feedback and those comments that have helped to improve our manuscript. We answer the comments point-by-point below:

**Comment #1:**

"DEMS is designed to conduct in-situ measurements of rapidly evolving aerosol systems" is slightly ambiguous here. The DEMS is indeed an advanced technique for probing the aerosol kinetic processes (e.g., evaporation and/or condensation), and this process should be within the classification region. However, it is not designed to handle rapidly evolving aerosols in the atmosphere.

**Response:**

Thanks for your suggestion. We have revised the sentence as follows: "DEMS is designed to conduct in-situ measurements of the kinetics of rapid aerosol processes happening inside its classification region, occurring at a timescale close to that of one EMS voltage scanning cycle".

**Comment #2:**

To better understand the dominating factor of the uncertainties of DEMS, I recommend the authors provide the uncertainty analysis in a more quantitative way, e.g., from the simulation perspective, such that the readers can directly compare uncertainties of the vapor pressure raised from various factors. Currently, only the uncertainty of diffusion coefficient is taken into account, and it would be better to demonstrate the uncertainties in numbers (e.g., put in a table). Other potential factors, including the assumption of constant Knudsen number, Kelvin effect, and particle residence time may also be considered for this estimation. Note as particles continue to shrink to a fairly small size, Brownian diffusion may also raise uncertainties, and this should be considered for future applications to nano-sized particles.

**Response:**

Thank you for your suggestions. Quantifying uncertainties is crucial for ensuring the robustness of our approach. In addition to diffusion coefficients, we have addressed uncertainties for several other factors in the current manuscript. For instance, the assumptions of constant Knudsen number and Kelvin effect have been examined through numerical simulations that did not rely on these

assumptions. The agreement between the theoretically derived and numerically obtained evaporation profiles (Fig. 3 in the main text) supports the validity of these assumptions. Other factors, such as the temperature gradient, vapor concentration gradient, and the exact particle evaporation time/residence time inside the classification region, indeed require further validation. These will be addressed through computational fluid dynamics and heat transfer simulations of the flow and temperature fields in our future work. Thank you for your note on the importance of Brownian Diffusion when measuring small-sized nanoparticles. This is another significant factor. We have added the following text to the manuscript to clarify these points:

Pg. 19: "The DEMS uncertainty bars largely overlap with the literature values, indicating a good agreement between the values, and reinforcing the accuracy of the DEMS in measuring the flat surface vapor pressures of organic nanoparticles. It should be noted that the uncertainties for other factors, such as the temperature gradient, vapor concentration gradient, and the exact particle evaporation time/residence time inside the classification region, may require further in-depth quantification. These will be addressed through computational fluid dynamics and heat transfer simulations of the flow and temperature fields in our future work. Furthermore, the importance of Brownian Diffusion needs to be incorporated when applying the method to small-sized nanoparticles."

**Comment #3:**

Line #299: "values we found from" – values were found from?

**Response:**

Thank you for the note, and we have revised the text on line 299 as follows: "Glycerol vapor pressure literature values were found from the Chemical Engineering and Materials Research Information Center (CHERIC)".

**Comment #4:**

Line #323: "Fig. S4, Fig. S5: - Fig. S5, Fig. S6?

**Response:**

Thank you for the note. We have revised the text on line 323 as follows: "These tails were found in the measurements of glycerol and PEG4 at 303 K (Fig. S5, Fig. S6)."